# Impact of COVID-19 Pandemic on Fragility Fractures of the Hip: An Interrupted Time-Series Analysis of the Lockdown Periods in Western Greece and Review of the Literature

**DOI:** 10.3390/geriatrics8040072

**Published:** 2023-07-01

**Authors:** Ilias D. Iliopoulos, Ioanna Lianou, Angelos Kaspiris, Dimitrios Ntourantonis, Christine Arachoviti, Christos P. Zafeiris, George I. Lambrou, Efstathios Chronopoulos

**Affiliations:** 1Laboratory for Research of the Musculoskeletal System, School of Medicine, National and Kapodistrian University of Athens, 10679 Athens, Greece; 2Department of Orthopaedic Surgery, “Rion” University Hospital and Medical School, School of Health Sciences, University of Patras, 26504 Patras, Greece; 3Accident and Emergency Department, “Rion” University Hospital and Medical School, School of Health Sciences, University of Patras, 26504 Patras, Greece; 4Orthopedics Department, General Hospital of Patras “Agios Andreas”, 26332 Patras, Greece; 5Choremeio Research Laboratory, First Department of Pediatrics, National and Kapodistrian University of Athens, 11527 Athens, Greece

**Keywords:** COVID-19, coronavirus, hip fracture, pandemic, lockdown, surgical timing, discharge

## Abstract

Since December 2019, the COVID-19 pandemic has had a significant impact on healthcare systems worldwide, prompting policymakers to implement measures of isolation and eventually adopt strict national lockdowns, which affected mobility, healthcare-seeking behavior, and services, in an unprecedented manner. This study aimed to analyze the effects of these lockdowns on hip-fracture epidemiology and care services, compared to nonpandemic periods in previous years. We retrospectively collected data from electronic patient records of two major hospitals in Western Greece and included patients who suffered a fragility hip fracture and were admitted during the two 5-week lockdown periods in 2020, compared to time-matched patients from 2017–2019. The results showed a drop in hip-fracture incidence, which varied among hospitals and lockdown periods, and conflicting impacts on time to surgery, time to discharge after surgery, and total hospitalization time. The study also found that differences between the two differently organized units were exaggerated during the COVID-19 lockdown periods, highlighting the impact of compliance with social-distancing measures and the reallocation of resources on the quality of healthcare services. Further research is needed to fully understand the specific variations and patterns of geriatric hip-fracture care during emergency health crises characterized by limited resources and behavioral changes.

## 1. Introduction

As of 1 March 2020, the outbreak of coronavirus disease 2019 (COVID-19) was characterized as a world pandemic by the World Health Organization (WHO). COVID-19, which is caused by a novel acute respiratory syndrome coronavirus, was first reported in December 2019, in Wuhan, Hubei, China [1]. Clusters of patients with pneumonia of unknown cause were reported and, since then, COVID-19 spread around the world and progressed into the second known pandemic of the 21st century [2,3,4,5]. This infection can result in respiratory, hepatic, gastrointestinal, and neuronal diseases [6,7]. Elderly patients are at high risk for serious complications, with overall mortality being five times greater for those older than 80 years old [8]. At the time of writing (April 2023), 6,887,000 deaths and 761,402,282 cumulative cases have been confirmed around the world [9].

On the healthcare-service front, the anticipated surge in COVID-19 cases led to significant organizational changes, including the redistribution of resources and manpower to address the needs arising from the pandemic [10]. Hospitals had to adapt and restructure their services, with many converting their main operating-theater area into COVID-19 critical care areas [11]. In addition, many hospitals had to cancel elective surgeries, while routine procedures became more complicated [12]. The fear of contracting the disease, coupled with the aforementioned changes, caused severe disruption of healthcare services leading to a reduction in patients’ access to emergency departments (EDs) and delays in diagnosis and treatment for non-COVID-19 related health issues, while the availability of rehabilitation services was also significantly restricted [13].

In addition to the changes in healthcare systems, the measures taken to control viral spread forced many people to stay at home, disrupting their daily routines and limiting social interactions, which caused a significant impact on people’s lifestyles, wellbeing, and psychological state [14,15,16,17]. Social distancing and isolation at home resulted in a sedentary lifestyle, muscular weakness, and impaired balance, which were reported to increase the risk of falls and concomitant hip fractures [18]. Similarly, elderly people living in residential homes presented high mortality rates during this period, as they were confined to their rooms with limited supervision, resulting in deterioration of their functional status and increased risk of falls [19]. One of the most common domestic injuries is hip fractures, with an estimated incidence of around 1.6% in Europe and 1.1% in the USA [20]. Hip fractures are notoriously associated with high mortality rates, residual disability, and serious socioeconomic implications [21]. They nearly always require hospitalization and surgical treatment and are fatal in almost a quarter of all cases [22,23,24]. The National Osteoporosis Guideline Group (NOGG) has reported that following a hip fracture, about half of those admitted can no longer live independently on discharge from the hospital, while only 30% of the patients fully recover [25].

The Greek population is the second oldest in Europe, after the Italian, with an estimated 21.5% of the people aged 65 years and over as of 2021, which puts them at higher risk for severe COVID-19 disease [26]. Given the limited financial capacity and resources of its healthcare system, the Greek government adopted a strategy that prioritized lockdowns and travel restrictions over massive population testing to mitigate the spread of the virus. Consequently, from 23 March 2020 to early May, and from 7 November 2020 to early January 2021, Greece implemented two massive national lockdowns [27].

The present study aims to analyze the impact of COVID-19 national lockdowns on fragility fractures of the hip, with respect to epidemiology and care service in two major public hospitals in Western Greece, the General Hospital of Patras (GHP) and the General Hospital of Aigio (GHA). Due to the fact that these institutions provide tertiary and secondary healthcare services, respectively. The study also aims to make comparisons and reach conclusions regarding the effects of lockdown measures on different trauma-level hospitals.

## 2. Materials and Methods

The present study followed the ethical principles of biomedical research, and it was approved by the Institutional Bioethics Board with registration number 40,297, complying with the 1964 Helsinki Declaration and its later amendment. This research received no funding.

We retrospectively analyzed the electronic patient health records of 2 general hospitals in Western Greece: the General Hospital of Patras “Agios Andreas” (GHP), and the General Hospital of Aigio (GHA). Both of these hospitals belong to the public healthcare system, are accessible to all Greek citizens, and provide primary trauma care along with continuous emergency-surgery services. GHA is a secondary healthcare institution located in the town of Aigio, which is situated in the northern part of the Peloponnese peninsula in Greece, and accepts emergency cases 24 h per day, 7 days per week. The hospital serves the wider Aigialeia municipality, which covers a rural area of approximately 729 square kilometers and includes several smaller towns and villages in addition to Aigio, with a population total of 46,990 people as of 2021 [28]. GHP is located in the city of Patras, the third-largest city in Greece (333.1 km^2^) with a population of 215,922 [28]. Along with the University Hospital of Rio in the city suburbs, it provides emergency services on a rotating basis and accepts referrals from the wider region of Western Greece providing tertiary care to patients with complex medical conditions.

We conducted an electronic database search of the medical records from GHA and GHP covering the two officially declared mandatory national lockdown periods in Greece, from 23 March 2020 to 27 April 2020, and from 7 November 2020 to 14 December 2020. All patients admitted to the hospitals’ emergency departments under the ICD-10 diagnostic codes of either S72.0 (femoral neck fracture) or S72.1 (pertrochanteric fracture) were included. Since the origin of the fracture was not specified in the database, we set an age threshold of 65 years and assumed that all fracture incidents above that age fall into the fragility fracture category. We collected demographic and clinical data for each patient, including age, gender, admission date, surgery date, and hospital-discharge date. To provide a basis for comparison with prepandemic, nonlockdown data, we retrieved identical patient information for the corresponding time frames of the previous three years (2017–2019) and grouped them to create the control cohort. We examined each lockdown period and each hospital separately in relation to the control and compared the number of cases, age, time to surgery, time to discharge after surgery, and total hospital stay. Additionally, in order to examine the differences in care service between a secondary and a tertiary healthcare institution and the possible effects of the lockdown, we compared the two hospitals side by side, combining the two separate time frames into a single nonlockdown period of 2017–2019 and a lockdown period of 2020.

All data was pseudonymized for the analysis and are openly available on reasonable request from the author. Data processing was performed using Origin Lab and Microsoft Excel and statistical analysis using the Mann–Whitney U Test with significance set at *p* ≤ 0.05. For statistically significant results, we measured effect size by calculating Cohen’s d and corrected the result using Hedge’s g to account for different sample sizes. All authors declare no conflicts of interest.

## 3. Results

### 3.1. Demographics and Fracture Incidence

There were no statistically significant differences concerning age or sex for the populations examined in both lockdown periods (Table 1 and Table 2).

For the first national lockdown period, from 23 March 2020 to 27 April 2020, our GHP electronic-database search identified eight suitable cases of hip fractures (six females). Correspondingly, a total of 43 patients (31 females) with hip fractures were admitted within the reference period of the three previous years (Table 1, Figure 1). For the same period, the GHA electronic database search revealed a 33% drop (n = 6 vs. n = 9) in fractures of the hip admitted to the hospital compared to the reference-control data from the previous three years (Table 1, Figure 1).

During the second lockdown period, 15 patients (12 females) with hip fractures were admitted to GHP. In relation to the reference data retrieved from the previous three years, this finding accounts for an 18% decrease in the number of hip fractures admitted (Table 2, Figure 1). Concerning GHA, our electronic-database search revealed a 40% decrease in the number of hip fractures admitted during the second lockdown period in relation to the reference years of 2017–2019 (n = 6 vs. n = 10) (Table 2, Figure 1).

### 3.2. Time to Surgery

During the first lockdown period, GHP data analysis revealed a 36% quicker time to surgery compared to the reference period, though the result did not reach statistical significance (*p* = 0.2) (Table 1, Figure 2). In GHA, the six patients identified by the electronic database search were operated on at a mean of 3.8 days (SD 1.8, median 4) compared to a mean of 4 days (SD 2.2, median 4) during the reference period (Table 1, Figure 2).

During the second lockdown period, retrieved data from GHP showed a 32% drop in time to surgery compared to the corresponding time period in 2017–2019, while in GHA, time to surgery was decreased by 22%, though both findings were not statistically significant (*p* = 0.147 and *p* = 0.433 respectively) (Table 2, Figure 2).

### 3.3. Time to Discharge after Surgery and Total Stay

In GHP, during the first lockdown period, we identified a 25% increase in time to discharge after surgery, though the result was not statistically significant (*p* = 0.359) (Table 1, Figure 3). 

On the other hand, the GHA database search and analysis identified a significant drop, of 42% (*p* = 0.034), in time to discharge after surgery during the first lockdown period of 2020 (Table 1, Figure 3). In regard to total stay, we found a 3% increase in the total time spent inside the hospital for the patients admitted to GHP, and a 29% decreased time for the patients of GHA during the lockdown period (*p* = 0.345 and *p* = 0.119 respectively) (Table 1, Figure 4).

During the second lockdown period, we recorded a reverse result for both hospitals in both times to discharge after surgery and total hospital stay (Table 2, Figure 3 and Figure 4)). In GHP, the data showed a 27% quicker time to discharge compared to the control (*p* = 0.268), while the total time spent inside the hospital also decreased by 29% (*p* = 0.271) (Table 2, Figure 3 and Figure 4). In GHA, however, the time to discharge after surgery increased by 57% (*p* = 0.072) and the total hospital stay time increased by 37% (*p* = 0.109) (Table 2, Figure 3 and Figure 4).

### 3.4. Comparing Secondary to Tertiary Healthcare

During the non-COVID reference period of 2017–2019, patients admitted to GHA with hip fractures experienced a significantly longer time to surgery by 19.9% (*p* = 0.031) compared to GHP (Table 3, Figure 5). 

This difference became even more pronounced during the lockdown periods, as data showed that patients of GHA needed 58.3% (*p* = 0.032) more time to reach the operating theater compared to the patients of GHP (Table 3, Figure 5). Similarly, the time to discharge after surgery and total hospital stay were also significantly longer in GHA compared to GHP during the reference period. These disparities were again exaggerated during the lockdown periods (Table 3, Figure 5).

## 4. Discussion

The devastating impact of COVID-19 in neighboring Italy prompted the Greek government to take swift action to prevent the virus from spreading in Greece [29]. On 9 March 2020, the Greek government implemented social-distancing measures which included closing schools and universities, banning public gatherings, and restricting travel [29]. These measures were soon followed by the first mandatory national lockdown on 23 March 2020, in order to protect the National Healthcare System [30]. The main slogan of this period across the country was “staying at home” and the priority was to avoid viral transmission in the elderly and vulnerable populations. After the first wave of the pandemic, in the summer of 2020, the government gradually began to ease restrictions and reopen the economy. However, this led to a second wave of COVID-19 cases in the fall of 2020, which resulted in the second full lockdown, which lasted from 7 November to 14 December 2020 [29]. 

Social distancing measures and the grade of compliance influenced the type of injuries mainly referred to in the ED throughout the periods of the two national lockdowns. During the first lockdown period, the incidence of hip fractures decreased in both hospitals with a recorded 44% decrease in GHP and a 33% decrease in GHA, compared to the control data from the grouped previous three years. Similarly, during the second lockdown period, the decrease in hip-fracture incidence in GHP and GHA was 18% and 40% respectively. These changes probably depict compliance with the restrictions, whereas no statistically significant changes regarding age or sex were noted. Even though, fragility fractures of the hip are mainly related to domestic accidents, the limited outdoor circulation may have influenced the results of this study, as also noted by Zhu et al., in a 2020 study of the fracture characteristics in the elderly [14]. The decline in the number of hip fractures in this study is also likely to have occurred due to the additional care provided by family members to elderly individuals, as the former stayed longer at home during the lockdown periods.

Several studies on the incidence of hip fractures, especially during the first lockdown, have been reported around the world (Table 4).

A decrease in the incidence of fragility fractures of the hip was reported in studies from many European countries [19,31,33,39,40,42], which mainly refer to periods corresponding to the first lockdown period in Greece. Similar results were noted by a multicenter study in Brazil which evaluated an extended period of social isolation [41], in a country with high compliance to the quarantine measures, in accordance with the situation in Greece during the first lockdown period. Conversely, several studies from the United Kingdom and the United States of America [32,34,35], reported an increase in the incidence of hip fractures during social-distancing periods, though other studies from those countries suggested that the rates remained relatively stable [36,37,38,43]. Research has shown that when elderly individuals are confined to their homes and engage in sedentary activities, they may be susceptible to an increased risk of falls and fractures [44]. In this respect, the increased incidence of hip fractures could be attributed to immobility resulting from COVID-19 lockdown measures, which, however, seems not to immediately affect the population of our study, even if the rate of decrease in the number of hip fractures in GHP seems to have slowed down during the second lockdown period. Overall, analysis of the literature reveals that the impact of COVID-19 social-distancing policies on the incidence of hip fractures can vary significantly, even within the same country or region, where a similar level of compliance can be assumed, taking into account the differences in demographics and the spread of the infection [42]. Furthermore, it appears that these effects may vary depending on the period of time studied, highlighting the complex interplay between public health measures and their impact on fracture rates.

Fragility fractures of the hip are accompanied by special perioperative needs which are imperative to be addressed in order to avoid high morbidity and mortality [45], including prompt operative treatment within 48 h [46]. However, achieving optimal patient-management times has always been a challenging process, whereas the COVID-19 pandemic has had a significant impact on theater productivity, adding further complexities to patient-care times. Interestingly, our study revealed a reduction in time to surgery during both the first and second national lockdown periods, for both hospitals examined, reaching up to 36% for GHP during the first lockdown. Although these findings did not reach statistical significance, they suggest a trend toward improved patient management during times of restricted hospital productivity. In regard to time to surgery, mixed results have been published in the literature with several authors reporting lower or comparable delays compared to the control groups from previous years [32,33,36,40,42,47,48], while others found an increase in time to surgery by up to 2.4 days by mean values [34,35,39,49] (Table 5).

Considering the reallocation of healthcare resources during the COVID-19 pandemic, the primary objective was to provide medical care to the maximum number of people while preventing potential viral transmission that could arise from surgical treatment delays [50]. The decrease in time to surgery, as reported by some authors, including the present study, exemplifies the efforts made to prioritize the urgent treatment of hip fractures [12]. This reduction in time to surgery also possibly highlights the benefits of quarantine measures for healthcare system capacity, as they effectively offloaded theaters by reducing surgical trauma, due to restricted mobility, and by canceling elective surgical cases. However, the productivity of the theatre, the allocation of surgical and anesthetic teams, and even the adaptation to complicated aerosol-generating procedures (AGPs) are factors that seem to have affected some hospitals, resulting in delays in operative treatment [34], while others have managed to overcome these challenges and maintain a high standard of healthcare services [38,50,51].

One of the challenges faced by hospitals worldwide during the COVID-19 pandemic was the imperative to minimize the risk of viral transmission within the hospital setting to both healthcare professionals and patients by minimizing both times to surgery and time to discharge after surgery. Several studies have reported mixed results on this topic, but focus only on total hospitalization time [19,32,33,35,39,41,52] (Table 6).

The present study examined both parameters and showed both to increase for GHP and a decrease for GHA during the first lockdown period, while a completely reversed effect was observed during the second lockdown period, though all results did not reach statistical significance. The tertiary hospital experienced a longer time to discharge, which can be attributed to the relocation of staff related to inpatient rehabilitation to COVID-19 critical-care units and the shortage of available beds at rehabilitation units during the first lockdown period. However, during the second lockdown, the GHP data seemed to reverse as the total number of COVID-19{+} cases spiked, and the need for available beds for COVID-19 patients implemented quicker inpatient rehabilitation and briefer total stays, despite limited resources. In contrast, the GHA seemed more influenced by the fear of in-hospital infection of elderly patients with fragility hip fractures and the higher motivation of more available staff in achieving shorter stays during the first lockdown period. However, during the second lockdown, these positive factors may have subsided and been seriously affected by the limitation of resources.

By analyzing the data collected for this study, we were able to make comparisons in relation to treatment standards for fragility hip fractures, between a tertiary (GHP) and a secondary (GHA) healthcare institution, during both the lockdown periods and normal non-COVID-19 time. Time to surgery, time to discharge, and total stay during the reference period of 2017–2019 were significantly longer in GHA than in GHP, and the differences were further amplified during the lockdown periods of 2020 (Table 3, Figure 5). These findings possibly highlight the benefit of treating such challenging injuries with high clinical and functional complexity in a tertiary healthcare setting, where a multidisciplinary approach can be implemented. Recent literature also emphasizes on orthogeriatric comanagement as the current gold standard of care for hip fractures, an approach that has been shown not only to improve timings but also decrease in-hospital complications, and in-hospital mortality compared to traditional care [53,54]. During the pandemic, the differences in delays within the hospital setting were more pronounced, possibly due to the greater caseload of COVID-19(+) patients in GHP, which imposed a need for quicker rehabilitation for hip-fracture patients to save more available beds. In-house COVID-19 testing was also performed in the tertiary hospital, limiting the time to surgery and total stay compared to the secondary hospital that relied on the capacity of other healthcare facilities for COVID-19 testing.

This study has several strengths and weaknesses. According to the known literature, this is the first study on the fragility hip-fracture incidence and treatment during both the first and the second national lockdown periods in Greece. Moreover, contrary to many studies using only a 1-year reference period to compare with 2020 COVID-19 data, we used data from a combined 3-year reference period to better stabilize the confounding weather conditions and strengthen our study. Additionally, this is the first time that data from patients with hip fractures in a tertiary and a secondary hospital of two towns in close proximity in Western Greece are compared and the standard of quality of services is evaluated [27,55,56].

Several limitations of this study include its retrospective design and the presentation of data from only two centers, both located in Western Greece. Furthermore, a significant limitation of this study is the narrow time frame that was examined, and the relatively small sample size of the populations studied. This may have been the primary reason why, in many cases, the results, although showing noticeable patterns, did not reach statistical significance. Some additional limitations that are commonly encountered in large administrative database studies can also be considered. First, there is a lack of specific clinical information on the fractures and hospital-discharge outcomes are limited to general measures such as age, sex, and diagnosis. Second, discharge summaries are produced by coding specialists, making them vulnerable to errors such as missing data and digitization errors. Finally, the database used in this study did not specify the origin of hip fractures, whether they were secondary to high-energy trauma or low-energy trauma, which we attempted to mitigate by introducing a lower age threshold of 65 years in accordance with the literature [57].

## 5. Conclusions

The findings of this study suggest that the COVID-19 pandemic contingency measures had an impact on the incidence of fragility hip fractures, which decreased during the study period. While some orthopedic services were disrupted to meet the demands of the pandemic, hip fractures should still be prioritized as they affect vulnerable elderly patients at risk of COVID-19 infection. The quality-of-care services depend on the capabilities and differences between hospitals but it seems possible to maintain a standard of care even with limited resources. The increased incidence of COVID-19 cases and deaths during the second lockdown, coupled with lower compliance with restrictions, further amplified the challenges faced by secondary healthcare institutions such as GHA compared to tertiary hospitals such as GHP, as evidenced by the study results. Therefore, during a pandemic crisis, contingency plans should prioritize not only the appropriate allocation of healthcare resources, equipment, and personnel but also the enhancement of the overall health of the geriatric population, with the goal of preventing both the immediate and long-term medical management of fragility fractures, amongst other conditions. Further similar multicenter studies are needed to identify specific patterns and variations in the care of geriatric hip fractures during times of emergency health crises that are marked by limited resources and behavioral changes.

## Figures and Tables

**Figure 1 geriatrics-08-00072-f001:**
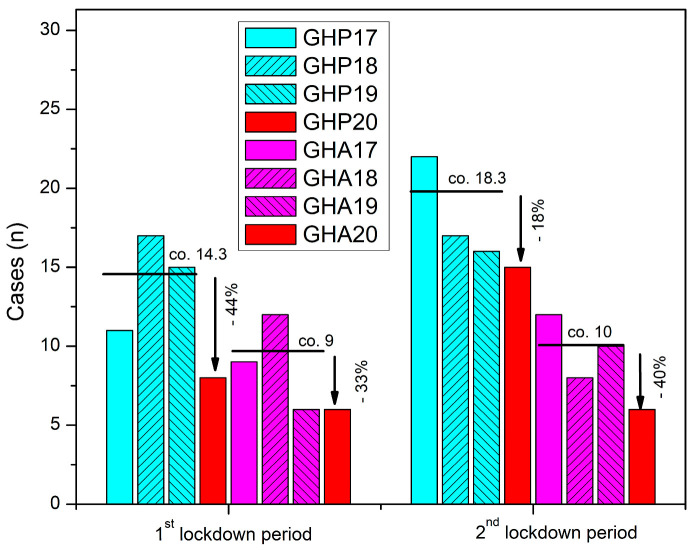
Number of patients with hip fracture admitted in GHP and GHA during the two national lockdown periods of 2020 compared to the previous three years (2017–2019).

**Figure 2 geriatrics-08-00072-f002:**
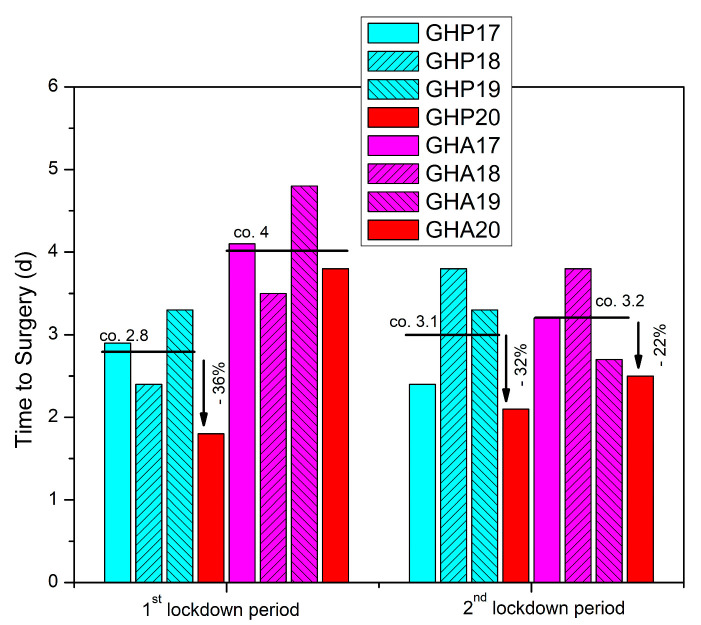
Time to surgery (d = days) for patients with hip fracture admitted in GHP and GHA during the two national lockdown periods of 2020 compared to the previous three years (2017–2019).

**Figure 3 geriatrics-08-00072-f003:**
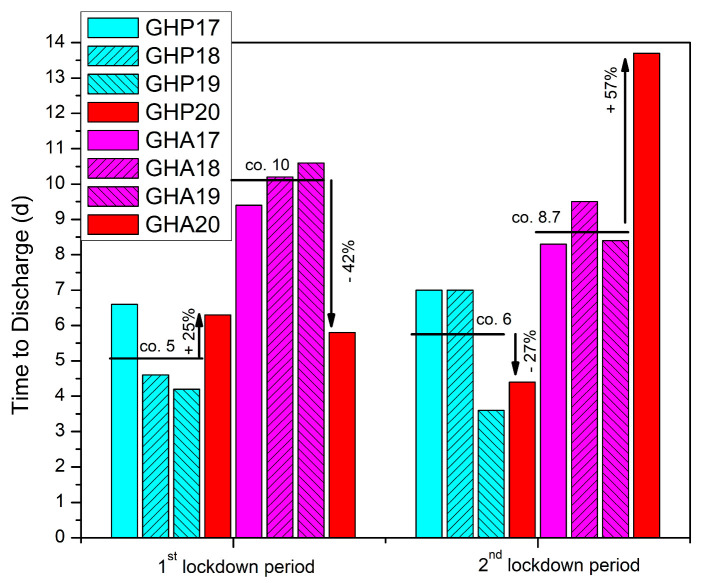
Time to discharge after surgery (d = days) for patients with hip fracture admitted in GHP and GHA during the two national lockdown periods of 2020 compared to the previous three years (2017–2019).

**Figure 4 geriatrics-08-00072-f004:**
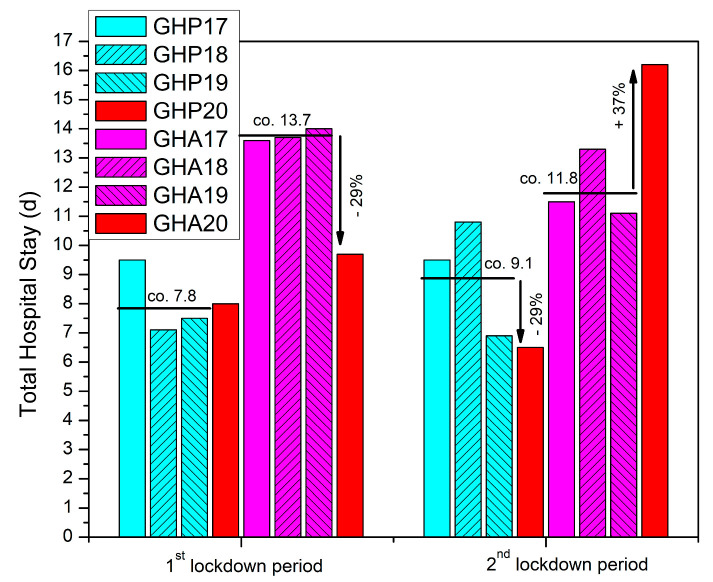
Total hospitalization time (d = days) for patients with hip fracture admitted in GHP and GHA during the two national lockdown periods of 2020 compared to the previous three years (2017–2019).

**Figure 5 geriatrics-08-00072-f005:**
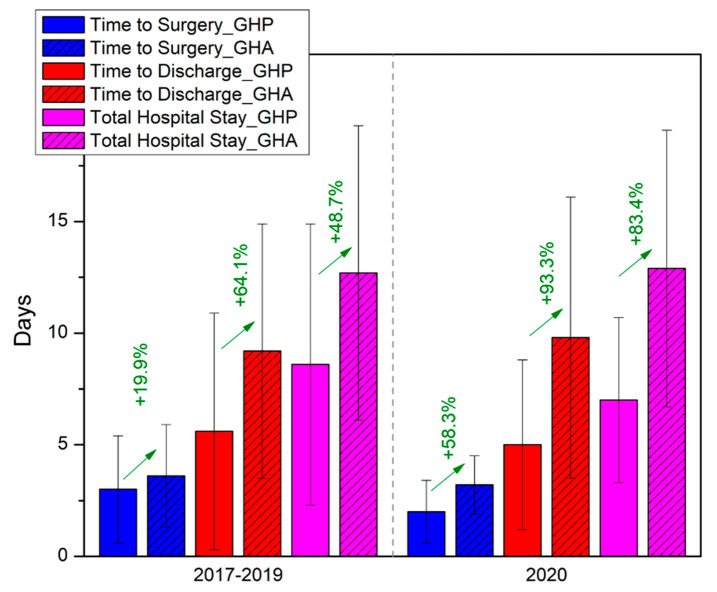
Side-by-side comparison between secondary (GHA) and tertiary (GHP) healthcare services and the impact of lockdown measures during the COVID-19 pandemic on the time parameters examined.

**Table 1 geriatrics-08-00072-t001:** Retrieved data and statistical analysis for the patients admitted with hip fracture (ICD-10: S72.0 or S72.1) during the corresponding time frame for the 1st lockdown period (20 March to 27 April).

		Grouped 2017–2019	2020	Difference	*p*
GHP					
Cases, n		14.3	8	−44%	
Age (y), mean		83.9	80.3	−3.6	0.127
	median	84	81.5		
	SD	7.8	6.9		
Time to Surgery (d), mean		2.8	1.8	−36%	0.200
	median	2	1		
	SD	2.4	1.1		
Time to Discharge (d), mean		5.0	6.3	+25%	0.359
	median	4	4.5		
	SD	3.5	6		
Total Hospital Stay (d), mean		7.8	8	+3%	0.345
	median	7	6		
	SD	4.2	5.8		
GHA					
Cases, n		9.0	6	−33%	
Age (y), mean		87.1	85.5	−1.6	0.303
	median	88	84		
	SD	7.7	8.8		
Time to Surgery (d), mean		4.0	3.8	−5%	0.697
	median	4	4		
	SD	2.2	1.3		
Time to Discharge (d), mean		10.0	5.8	−42%	0.034
	median	7	6		[g = 0.7]
	SD	6.8	2.0		
Total Hospital Stay (d), mean		13.7	9.7	−29%	0.119
	median	11	10.5		
	SD	7.9	2.8		

Abbreviations: n = number, y = years, d = days, SD = standard deviation, GHP = General Hospital of Patras, GHA = General Hospital of Aigio, g = Hedge’s g.

**Table 2 geriatrics-08-00072-t002:** Retrieved data and statistical analysis for the patients admitted with hip fracture (ICD-10: S72.0 or S72.1) during the corresponding time frame for the 2nd lockdown period (7 November to 14 December).

		Grouped 2017–2019	2020	Difference	*p*
GHP					
Cases, n		18.3	15	−18%	
Age (y), mean		82.5	86.1	+3.6	0.061
	median	84	88		
	SD	6.6	7.8		
Time to Surgery (d), mean		3.1	2.1	-32%	0.147
	median	2	2		
	SD	2.4	1.5		
Time to Discharge (d), mean		6.0	4.4	−27%	0.268
	median	4	4		
	SD	6.3	1.1		
Total Hospital Stay (d), mean		9.1	6.5	−29%	0.271
	median	7	6		
	SD	7.5	1.5		
GHA					
Cases, n		10.0	6	−40%	
Age (y), mean		85.4	83.8	−1.6	0.305
	median	86	84.5		
	SD	7.1	7.3		
Time to Surgery (d), mean		3.2	2.5	−22%	0.433
	median	2	2.5		
	SD	2.2	1		
Time to Discharge (d), mean		8.7	13.7	+57%	0.072
	median	7.5	14.5		
	SD	4.5	6.7		
Total Hospital Stay (d), mean		11.8	16.2	+37%	0.109
	median	10	18		
	SD	5.2	6.9		

Abbreviations: n = number, y = years, d = days, SD = standard deviation, GHP = General Hospital of Patras, GHA = General Hospital of Aigio.

**Table 3 geriatrics-08-00072-t003:** GHP and GHA data from the two corresponding time periods of national lockdowns combined into a single COVID-19/lockdown 2020 cohort and a non-COVID-19/lockdown cohort of the grouped previous three years (2017–2019) and statistical analysis.

9	Non-COVID/Lockdown Period (2017–2019)					COVID/Lockdown Period (2020)			
		GHP	GHA	Difference	*p*	GHP	GHA	Difference	*p*
Cases, n		98	57			23	12		
Age (y), mean		83.1	86.2	+3.1 y	0.123	84.0	84.7	+0.7 y	0.976
	median	84	87			87	84		
	SD	7.2	7.4			8.0	8.1		
Time to Surgery (d), mean		3.0	3.6	+19.9%	0.031	2.0	3.2	+58.3%	0.032
	median	2	3		[g = 0.4]	2	3.5		[g = 0.8]
	SD	2.4	2.3			1.4	1.3		
Time to Discharge (d), mean		5.6	9.2	+64.1%	<0.001	5.0	9.8	+93.3%	0.007
	median	4	7		[g = 1.1]	4	7		[g = 1.0]
	SD	5.3	5.7			3.8	6.3		
Total Hospital Stay (d), mean		8.6	12.7	+48.7%	<0.001	7.0	12.9	+83.4%	0.004
	median	7	11		[g = 1.0]	6	11.5		[g = 1.1]
	SD	6.3	6.6			3.7	6.2		

Abbreviations: n = number, y = years, d = days, SD = standard deviation, GHP = General Hospital of Patras, GHA = General Hospital of Aigio, g = Hedge’s g.

**Table 4 geriatrics-08-00072-t004:** Summary of available studies examining hip fracture incidence changes during COVID-19 social-distancing measures.

Study	Country	Type of Study	Period of Study	Scope	Incidence of Hip Fractures	Comments
Maniscalo et al., 2020, [31]	Italy	Retrospective	22/02/20–18/04/20 (rp: 23/02/19–20/04/19)	Local	↓	Relative ↓
Egol et al., 2020, [32]	United States of America	Prospective	01/02/20–15/04/20 (rp: same in 2019)	Regional	↑	138 hip fxs in 2020, 115 in 2019
Malik-Tabassum, et al., 2020, [33]	United Kingdom	Retrospective	23/03/20–11/05/20 (rp: same in 2018, 2019)	Local	↓	General ↓ in trauma workflow, marked ↓in younger adults requiring surgery
Narang et al., 2020, [34]	United Kingdom	Prospective	01/03/20–30/04/20 (rp: same in 2019)	Regional	↑	682 hip fxs in 2020, 664 in 2019
Arafa et al., 2020, [35]	United Kingdom	Retrospective	01/03/20–31/05/20 (rp: same in 2019)	Local	↑	61.7% increase compared with 2019
Hall et al., 2020, [36]	United Kingdom (Scotland)	Retrospective	01/03/20–15/04/20	National	→	Differences between regions due to testing strategies, social distancing policies, and ld measures.
Ogliari et al., 2020, [37]	United Kingdom	Prospective	1st–12th w 2020 (pre-ld), 13th–19th w 2020 (ld) (rp: same in 2015–2019)	Local	→	Hip fxs unchanged during ld, ↓ in nonhip fragility fxs
Haskel et al., 2020, [38]	United States of America	Retrospective	22/03/20–30/04/20 (rp: same in 2019)	Local	→	overall ↓ in ortho cases
Ojeda-Thies et al., 2020, [19]	Spain	Retrospective	01/03/20–01/05/20 (rp: same in 2018, 2019)	Local	↓	5.5% of pts presented >24 h after injury
Wignall et al., 2021, [39]	United Kingdom	Retrospective	01/03/20–30/05/20 (rp: same in 2019)	Regional	↓	↓ admissions in total (304 vs. 276). → average admissions/d.
Crozier-Shaw et al., 2021, [40]	Ireland	Retrospective	20/0320–20/05/20 (rp: same in 2019)	Regional	↓	20% ↓ in hip fxs
Da Silva et al., 2022, [41]	Brazil	Retrospective	03/20–12/20(rp: same in 2019)	National and regional	↓	↓ in domestic accidents
Troiano et al., 2023, [42]	Italy	Retrospective	09/03/2020–03/05/2020, (rp: same in 2019, 2021)	Regional	↓	Almost unchanged (36 pts in 2019 vs. 28 in 2020 vs. 29 in 2021)

Symbols/Abbreviations: ↓ = decrease, ↑ = increase, → = stable, rp = reference period, ld = lockdown, fxs = fractures, pts = patients.

**Table 5 geriatrics-08-00072-t005:** Summary of available studies examining delays in surgical treatment of hip fractures during COVID-19 social-distancing measures.

Study	Country	Type of Study	Period of Study	Scope	Time to Surgery	Comments
Egol et al., 2020, [32]	United States of America	Prospective	01/02/2020–15/04/2020 (rp: same in 2019)	Regional	→	↑ for COVID-19{+} pts vs. COVID-19{−} pts
Malik-Tabassum et al., 2020, [33]	United Kingdom	Retrospective	23/03/2020–11/05/2020 (rp: same in 2018, 2019)	Local	↓	Not statistically significant
Muñoz Vives et al., 2020, [49]	Spain	Retrospective	14/05/2020–04/03/2020 (no rp)	National	→	Results similar to the Spanish National Hip Fracture Registry
Narang et al., 2020, [34]	United Kingdom	Prospective	01/03/2020–30/04/2020 (rp: same in 2019)	Regional	↑	Delays due to complex AGPs,
Dupley et al., 2020, [48]	United Kingdom	Retrospective	01/03/2020–26/04/2020 (rp: same in 2016–2019)	National	↓	In 64% definitive surgery within 36 h, vs. 60.1% and 58% in March and April 2016–2019
Arafa et al., 2020, [35]	United Kingdom	Retrospective	01/03/2020–31/05/2020 (rp: same in 2019)	Local	↑	Not statistically significant
Hall et al., 2020, [36]	United Kingdom (Scotland)	Retrospective	01/03/2020–14/04/2020	National	→	No difference when compared 23 d pre and after ld
Ojeda-Thies et al., 2021, [47]	Spain	Retrospective	01/03/2020–01/05/2020 (rp: same in 2018, 2019)	Local	↓	Controversial results, authors avoided surgery upon systematic inflammatory involvement
Wignall et al., 2021, [39]	United Kingdom	Retrospective	01/03/2020–30/05/2020 (rp: same in 2019)	Regional	↑	COVID-19{+} pts with significantly higher delay compared to COVID-19{−}
Crozier-Shaw et al., 2021, [40]	Ireland	Retrospective	20/03/2020–20/05/2020 (rp: same in 2019)	Local	↓	57.8% of the pts treated within 48 h in 2019, 78% in 2020
Troiano et al., 2023 [42]	Italy	Retrospective	09/03/2020–03/05/2020 (rp: same in 2019, 2021)	Regional	→	No statistically significant change during ld, longer in 2021 group

Symbols/Abbreviations: ↓ = shorter time, ↑ = longer time, → = no difference, rp = reference period, ld = lockdown, pts = patients, d = days, h = hours, AGPs = aerosol-generating procedures.

**Table 6 geriatrics-08-00072-t006:** Summary of available studies examining the impact of COVID-19 social-distancing measures in total hospitalization time of patients presenting with hip fracture.

Study	Country	Type of Study	Period of Study	Scope	Length of Stay	Comments
Egol et al., 2020, [32]	United Kingdom	Prospective	01/02/2020–15/04/2020 (rp: same in 2019)	Regional	↑	COVID-19{+} and pts suspected for infection with greater length of stay than COVID-19{−} pts
Malik-Tabassum et al., 2020, [33]	United Kingdom	Retrospective	23/03/2020–11/05/2020 (rp: same in 2018, 2019)	Local	↓	Higher portion of pts discharged to rehabilitation center
Arafa et al., 2020, [35]	United Kingdom	Retrospective	01/03/2020–31/05/2020 (rp: same in 2019)	Local	Variable	Mean total length of stay significantly longer in COVID-19{+} pts compared to COVID-19{−} pts and 2019 pts
Hall et al., 2020, [36]	United Kingdom (Scotland)	Retrospective	01/03/2020–15/04/2020	National	↓	Significant ↓ due to fear of viral transmission, or more resources available from ↓in polytrauma pts
Ojeda-Thies et al., 2021, [47]	Spain	Retrospective	01/03/2020–01/05/2020 (rp: same in 2018, 2019)	Local	↓	↓length of stay in pre COVID-19 period and COVID-19{−} pts
Shemesh et al., 2021, [52]	Israel	Retrospective	01/03/2020–31/05/2020 (rp: same in 2017, 2018)	Local	↓	Influenced by the fear of in-hospital viral transmission
Wignall et al., 2021, [39]	United Kingdom	Retrospective	01/03/2020–30/05/2020 (rp: same in 2019)	Regional	↓	Significant ↑in pre COVID-19 group and COVID-19{+} pts compared to COVID-19{−}
da Silva et al., 2022, [41]	Brazil	Retrospective	03/2020–12/2020 (rp: same in 2019)	National and regional	↓	Reflects the need for beds for COVID-19(+) pts

Symbols/Abbreviations: ↓ = decrease, ↑ = increase, rp = reference period, pts = patients.

## Data Availability

Data are openly available on reasonable request from the corresponding authors.

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
