# Peer review of "Impact of COVID-19 Pandemic on Fragility Fractures of the Hip: An Interrupted Time-Series Analysis of the Lockdown Periods in Western Greece and Review of the Literature"

_geriatrics, 2023, doi:10.3390/geriatrics8040072_

Round 1

Reviewer 1 Report

This is a very comprehensive piece of work, though perhaps rather long considering the small number of patients on which it is based. It serves as an absolutely excellent review of the pandemic from the perspective of hip fracture, but the presentation of the patient data could be made much more concise, so that this data serves more as an illustration of themes detected in the literature review, which would avoid the need for quite so many graphs, tables and statistics.

The literature review seems to be focused on academic papers, but doesn't reference work on huge numbers of patients published by national clinical audits of hip fracture in their annual reports. The biggest COVID-19 study was based on national clinical audit data (doi:10.1302/0301-620X.103B6. BJJ-2021-0326) but appears to be missing from the references.

Author Response

Dear Reviewer,

Thank you for your time and effort to thoroughly review our manuscript. We appreciate your positive feedback on the comprehensive nature of our work and its contribution as a review of the pandemic's impact on hip fractures. We acknowledge your suggestions towards a shorter and more concise presentation and we addressed them as follows:

Point 1: Manuscript rather long considering the small number of patients on which it is based.

Response 1: We carefully revised the whole manuscript removing non-relevant infos, unnecessary words and sentences and compacting the whole text, especially the introduction part down to less than 1 page. Concerning the tables in the Discussion part, we changed dates to numbering format, we used more symbols and abbreviations and removed unnecessary infos form the comments column to have them more compacted and readable.

Point 2: Presentation of the patient data could be made much more concise.

Response 2: We carefully revised the Results part shortening the sentences and avoiding presenting too manny numbers since they are also shown in tables and figures. We also acknowledge the need to unload our tables of too many data and we removed the columns reporting data for each of the previous three years (2017-2019).

Point 3: The literature review doesn't reference work on huge numbers of patients published by national clinical audits of hip fracture in their annual reports (doi: 10.1302/0301-620X.103B6. BJJ-2021-0326 missing from the references).

Response 3: We added the suggested paper by Johansen and Inman in our references (Ref 43). We also added the study of Troiano et al. and included it in our comparisons table (ref 42).

Reviewer 2 Report

Dear Authors: I want to congratulate with You for Your work, and for the paper. It is very interesting and it deserves to be published but, in my opinion, some revision is needed. First of all, the paper is too long, especially in some chapters.

Introduction: definitely too long. please remove all non relevant infos and compact all the others. intro should be (less than) 1 page.

Materials and methods: too long. Please remove all word and sentences that are not necessary ("strictly" in line 145, "and are funded by taxes" in line 152, "According to the latest available data from the Greek National Statistical Service" in line 158-159, and so on..)

Results: again, too long. try and shorten the sentences in which You present Your results, being repeated in table and figures as well. Also, I cannot clearly understand why to present data for each years and aggregate data as well for the pre-covid period.

Discussion: too long, try and shorten sentences, try and remove nonrelevant infos and comments. On the other hand, Your comparison to other studies is very interesting but You miss the one most close to Yours as for study design (Troiano et al. Impact of COVID-19 Pandemic on Treatment and Outcome of Fragility Hip Fractures In Non-COVID Patients: Comparison Between the Lockdown Period, a Historical Series and the "Pandemic Normality" in a Single Institution. GOS J 2023 doi: 10.1177/21514593231152420): You should include this study in Your comparison. As for tables, with the help of the editorial office, try and evaluate if presenting them differently (upright?, more symbols, dates in numbers?) to have them more compacted and readable.

Conclusions: should be more general (or generalizable).

Bibliography: Ref 36 and 37 should be written with latin alphabet, other than the above mentioned paper should be included.

Minor editing is required

Author Response

Dear Reviewer,

Thank you for your kind words and congratulations on our work. We appreciate the time and effort you have dedicated to providing valuable feedback and suggestions for improving our work. Your insights have greatly contributed to enhancing the clarity and rigor of our work and we tried to carefully address each comment as follows:

Point 1: Introduction definitely too long. please remove all non relevant infos and compact all the others. intro should be (less than) 1 page.

Response 1: We carefully revised this part removing non-relevant infos and compacting the whole text down to less than 1 page. 

Point 2: Materials and methods too long. Please remove all word and sentences that are not necessary.

Response 2: We carefully revised this part removing all unnecessary words and sentences as per your suggestions.

Point 3: Results too long. try and shorten the sentences in which you present your results, being repeated in table and figures as well.

Response 2: We carefully revised this part shortening the sentences and avoiding presenting too manny numbers since they are also shown in tables and figures.

Point 4: I cannot clearly understand why to present data for each years and aggregate data as well for the pre-covid period.

Response 4: We acknowledge the need to unload our tables of too many data and we removed the columns reporting data for each of the previous three years (2017-2019).  

Point 5: Discussion too long, try and shorten sentences, try and remove non-relevant infos and comments.

Response 5: We carefully revised this part removing all non-relevant infos and shortening sentences.

Point 6: You miss the study most close to yours as for study design (Troiano et al. Impact of COVID-19 Pandemic on Treatment and Outcome of Fragility Hip Fractures In Non-COVID Patients: Comparison Between the Lockdown Period, a Historical Series and the "Pandemic Normality" in a Single Institution. GOS J 2023 doi: 10.1177/21514593231152420): You should include this study in your comparison.

Response 6: We cited the suggested study and included it in our comparisons tables.

Point 7: As for tables, with the help of the editorial office, try and evaluate if presenting them differently (upright?, more symbols, dates in numbers?) to have them more compacted and readable.

Response 7: We carefully revised the tables in the Discussion part. We changed dates to numbering format, we used more symbols and abbreviations and removed unnecessary infos form the comments column.

Point 8: Conclusions should be more general (or generalizable).

Response 8: We slightly revised this part to make it more generalizable.

Point 9: Bibliography Ref 36 and 37 should be written with latin alphabet, other than the above mentioned paper should be included.

Response 9: We removed greek alphabet from refs 36,37 (28,29 in the revised text) and added the suggested paper from Troiano et al (ref 42).